# Accidental Injury or “Shaken Elderly Syndrome”? Insights from a Case Report

**DOI:** 10.3390/healthcare11020228

**Published:** 2023-01-12

**Authors:** Valentina Bugelli, Carlo Pietro Campobasso, Alessandro Feola, Ilaria Tarozzi, Arturo Abbruzzese, Marco Di Paolo

**Affiliations:** 1South-East Tuscany Local Health Agency, 58100 Grosseto, Italy; 2Department of Experimental Medicine, University of Campania “Luigi Vanvitelli”, Via Luciano Armanni 5, 80138 Naples, Italy; 3Modena Local Health Agency, 41126 Modena, Italy; 4Neuroradiology Unit, Azienda Ospedaliero-Universitaria Pisana-Santa Chiara, 56100 Pisa, Italy; 5Department of Surgical Pathology, Medical, Molecular and Critical Area, Institute of Legal Medicine, University of Pisa, 56126 Pisa, Italy

**Keywords:** elder maltreatment, elder abuse, subdural haemorrhage, traumatic brain injury, shaking

## Abstract

Subdural haemorrhage (SDH) as result of a traumatic brain injury (TBI) is a common cause of death in cases of fatal physical abuse. Since intracranial bleeding is a common finding in elderly due to age-related intracranial changes or increasing prevalence of anticoagulant medication, differential diagnosis between inflicted and non-inflicted head injury is challenging. A case of an elderly woman’s death caused by TBI is reported. Autopsy showed multiple polychromatic bruises and a frontoparietal hematoma with bilateral subacute SDH. History excluded paraphysiological or pathological non-traumatic conditions that could justify SDH, while iatrogenic factors only played a contributory role. Since polychromatic bruises distributed on the face, the upper extremities and the chest were consistent with forceful grasping/gripping or repeated blows and SDH can form in absence of impact or by mild/minor blows, SDH was considered the result of repeated physical abuses. Differential diagnosis between traumatic and non-traumatic SDH is still challenging for forensic pathologists. As largely accepted in the pediatric population and occasionally described also in adults, however, violent shaking should be also considered as a possible mechanism of SDH—especially in elderly who do not have any sign of impact to the head.

## 1. Introduction

In most countries, 65 is commonly the age of eligibility for retirement and old-age social programs and, therefore, this is the age at which individuals are mostly considered elderly [1,2]. In the near future, global population aging will significantly increase the number of older people—especially those who are care-dependent [3]. It is estimated that many low- and middle-income countries will experience sharper increases in population aging and, in some cases, a doubling in the absolute number of older people by 2050. At least 10% of adults age 65 and older are expected to experience some form of elder abuse [4]. According to recent reports, during the COVID-19 pandemic, this phenomenon has increased further, with a heavy burden on older people and their caregivers [5,6,7,8].

Elder abuse, also called elder mistreatment or elder maltreatment, is a violation of human rights. Article 25 of the Charter of Fundamental Rights of the European Union (adopted in Nice on 7 December 2000) recognizes and respects the rights of older people to lead a life of dignity and independence and to participate in social and cultural life. Elder abuse is defined by the World Health Organization (WHO) as a single or repeated act, or a lack of appropriate action, occurring within any relationship where there is an expectation of trust, which causes harm or distress to an older person [4].

It includes different categories: physical abuse—that is, a violent act carried out with the intention to cause physical pain or injury to an older adult; sexual abuse, defined as nonconsensual sexual contact of any kind; psychological or verbal abuse—that is, a violent non-physical act intended to cause emotional pain to the elder; financial exploitation, defined as misappropriation of an older person’s money or property; neglect—that is, intentional or unintentional failure by a caregiver (active or passive caregiver neglect) or oneself (self-neglect) to provide the elderly with proper care. The National Aging Resource Center on Elder Abuse (NARCEA) also comprises a miscellaneous group including violation of an individual’s rights, such as exclusion from decision-making processes dealing with health, personal issues and marriage [9,10,11].

Elder maltreatment is a widespread, though often underestimated, phenomenon [2,3,4,5,6,7,8,9,10,11,12,13]. The prevalence of this phenomenon in institutions such as hospitals, nursing homes and other long-term care facilities is estimated to be alarmingly high, reaching 64.2% in some studies [2,14]. The close relationship between the victims and the caregivers is also the main reason why only 4% of elder abuse is reported. Therefore, elder abuse is a worldwide hidden and under-reported problem, although it is estimated that 1 in 6 older adults have been abused in the past year [4,15,16,17,18]. According to the meta-analysis performed by Yon et al. (2017), 15.7% of people aged 60 and older in the general population were victims of some form of abuse [19]. Even higher rates are reported in studies conducted in some specific community settings, such as in Indian slum regions [20].

Among the additional reasons for under-reporting, there are also problems of identifying elder abuse. Elder adults are in fact frequently reluctant to tell of the mistreatment they suffer. A lack of education and formal training among healthcare workers in recognizing abuse and the protocols for reporting abuse to judicial authorities as required by law have also been mentioned. However, one of the main reasons is also the following: the identification of abuse is mostly an uncertain diagnosis, which increases practitioner’s fear that they may do more harm by taking action.

Kleinschmidt in particular emphasized that physicians often have fear of offending patients and their families [17]. Clarke et al. underlined that physicians are often concerned about retaliatory litigation from the elderly or their relatives [21]. In 2018, the European Council of Legal Medicine (ECLM) has recommended the guidelines for the examination of suspected elder abuse with the purpose of providing a common framework for healthcare professionals and forensic practitioners to properly document and evaluate potential victims of elder abuse [22].

Physical abuse and neglect are forms of abuse that are frequently seen among deaths from elder maltreatment [23].

Physical signs of abuse may overlap with the symptoms and outcomes of various diseases or side effects of medications [9,10,11]. For examples, bruises may be due to high doses of anticoagulants or they may be misclassified as senile purpura [24,25]. Since elderly people are commonly affected by multiple diseases as well as aging process-related conditions, the consequences of physical abuse can be devastating and can be difficult to distinguish from the accidental injuries that elderly people are more prone to than the general population. Head trauma is prevalent in cases of severe physical abuse, related to high morbidity and mortality rates, with subdural haemorrhages commonly found as causes of death in cases of elder abuse [26,27].

Unfortunately, ischemic strokes and non-traumatic intracranial bleeding including subdural hematomas, intraparenchymal hemorrhages, subarachnoid hemorrhages may mimic the effects of blunt trauma, with similar symptoms such as headache, altered mental status or focal neurological deficits [28,29,30,31]. Moreover, intracranial changes that occur with aging (e.g., increased adherence of the dura to the skull, cerebrovascular atherosclerosis increasing bridging vein fragility and progressive brain atrophy), as well as the increasing prevalence of anticoagulant medication use, puts older adults at risk of intracranial bleeding—even after mild head trauma [32,33].

We report a case of an elderly woman’s death caused by TBI and the difficulties related to differential diagnosis between inflicted and non-inflicted traumatic injuries in elderly people in judicial cases.

## 2. Case Report

A 82-year-old Caucasian woman living with her husband and a caretaker was found at home unconscious and unresponsive. The caretaker said the old woman had been put to bed about two and a half hours before. Soon after the reported finding, the woman was transported to a third level hospital’s Emergency Department. A medical history of widespread cortical atrophy, progressive cognitive impairment, depression, hypertension, hypercholesterolemia, osteoporosis, megaloblastic anemia and mild renal insufficiency was reported. The therapy adopted before the hospitalization was represented by antidepressants such as trazodone, medications for treating high blood pressure such as anti-calcium channel blockers (amlodipine) and platelet aggregation inhibitors such as ticlopidine. A cranioencephalic CT-scan showed a right parieto-temporal subdural hematoma (SDH), 21 mm thick, compressing the right lateral ventricle and third ventricle, causing a midline shift of about 9 mm. Additional findings were represented by a left temporo-occipital SDH, 8 mm thick, right temporo-basal intraparenchymal hemorrhages and blood in the fourth ventricle (Figure 1A). No fractures or other traumatic injuries of the internal organs were found in the thorax or abdomen. There was no indication of neurosurgical treatment. It is worth mentioning that a cranioencephalic CT-scan was performed just two months before the event. This previous imaging exam showed findings consistent with chronic ischemic encephalopathy and enlargement of intracranial cerebrospinal fluid spaces, but no other pathologic abnormalities were reported (Figure 1B).

According to the three aspects of responsiveness, a score of three was assessed on the Glasgow Coma Scale (GCS): no eye opening; no verbal response; no motor response. The GCS was permanently equal to three until brain death was pronounced, four days after unsuccessful intensive care. At a post-mortem examination, the patient was under-weight, with a Body Mass Index (BMI) of 18.47 kg/m^2^. She was 149 cm high and 41 kg in weight. Subcutaneous tissue was low. Multiple bruises were found on the head and face, neck, thorax and upper and lower extremities (Figure 2A–C).

Over the sacrum there was a small area of skin loss resembling a bedsore at a very early stage. In total, 56 small bruises associated with macroscopic bleeding into the skin were detected. Bruises of different sizes were represented, mostly oval/round in shape, ranging from 0.5 cm up to 6 cm in diameter, and from red through purple to yellow/green in color. Cranial bruises were mostly located on the vertex but also on the root of the nose and on the left side of the face (in particular, at the upper eyelid, at the posterior surface of the left ear and the cheek and, finally, under the chin). Multiple contusions were also distributed in the dorsal and lateral surfaces of the upper extremities and lateral breast on both sides, consistent with forceful grasping/gripping or repeated blows. At autopsy, a frontoparietal hematoma, 4 × 3.5 cm in dimension, was associated with a bilateral subacute SDH, approximately 2 cm thick, with fresh coagulated blood between the arachnoid and dura mater. No skull fractures and no traumatic injuries of the hyoid bone were found. Unfortunately, no ophthalmic examination was performed; neither the optic nerve nor the eyes were removed at autopsy. At the neck, no hemorrhages of the soft tissue were found. The heart showed coronary atherosclerosis with mild non-obstructive plaques. Toxicological analyses were negative for alcohol and common illicit and licit drugs in peripheral blood samples as well as in hematoma blood samples. Microscopic examination of bruises and SDHs was performed using basic histological stains such as hematoxylin-eosin. The main purpose of the histological examination was to search for tissue characteristics useful for determining the age of the brain and skin injuries. In fact, although bruises cannot be aged by their color, the color may enable one to suspect whether a skin injury is fresh or not fresh, to then be confirmed by conventional histology. According to the histological examination of the scalp, face, chest and upper and lower extremities, skin samples were collected and more than 10 bruises were assessed to be older than 5 days. These were from blue/purple to yellow/green in color and were located at the face, the breast and the upper limbs. The main microscopic findings were the detection of granulation tissue with new capillary blood vessels, dense collagen fibers with fibroblasts and scar tissue and the infiltration of macrophages and lymphocytes [34]. These histological findings were observed in samples collected from the head, the face, the lateral chest of both sides and the upper extremities (Figure 3).

The SDHs showed histological findings consistent with subacute hemorrhages older than 5 days but less than a week, as represented by the formation of granulation tissue with associated breakdown of erythrocytes and some siderophages. A thin layer of fibroblasts was observed between the dura and the clot, and the latter was invaded by new capillary blood vessels [34]. Intense edema was also present based in large pericellular and perivascular empty spaces (Figure 4).

Death was finally assessed to be due to a mechanical closed brain injury, represented by SDHs causing cerebral compression, in a frail elderly woman with cognitive impairment. Based on this assessment, the TBI was the result of a physical abuse by slapping, pushing, shaking and kicking according to the multiple bruises in different anatomical regions.

## 3. Discussion

TBI is a common cause of death in all age groups, mostly resulting from an impact suffered in traffic or sport accidents, accidental falls, violent blows delivered by hands, feet or different types of weapons (bottles, hammers, wooden bats, etc.) or objects falling on the head. However, closed brain injury can also be the result of a non-contact trauma associated with a violent head motion due to acceleration/deceleration and rotational movements, as occur in shaking. In violent shaking, rapid acceleration/deceleration movements of the skull in relation to the trunk are inflicted to an individual held by the upper extremities or the thoracic region. The mechanism of shaking as a significant mechanism of brain injury has been widely accepted in the pediatric population and is better known as “Shaken Baby Syndrome” (SBS), a definition that has been changed by a consensus statement to “Abusive Head Trauma” (AHT). The etiology of AHT is multifactorial, including shaking, shaking with impact, impact alone, etc. [33]. The classical triad of SBS (SDH, retinal hemorrhages and hypoxic encephalopathy) has been now enriched by a compilation of signs and symptoms including: irritability/lethargy, altered mental status, respiratory impairment, multiple fractures (especially when in different stages of healing), varying degrees of abrasions or bruising in unusual locations (especially when observed in a non-mobile child), vomiting and poor nutrition [34,35]. However, no single injury is diagnostic of AHT. Based on this knowledge and experience with the pediatric population, a similar mechanism of TBI has been proposed in adults and the elderly, in so-called “Shaken Adult Syndrome” (SAS). Four suspected cases of SAS have been reported in the literature. Pounder (1997) described the use of repeated shaking used by the Israeli General Security Service during the interrogation of a 30-year-old Palestinian male of 44.3 kg weight and 151 cm height. The victim was shaken 12 times over the course of the interrogation for many hours. The autopsy disclosed extensive pectoral and shoulder bruising, along with acute SDH and diffuse axonal injury consistent with violent shaking, but no other trauma [36,37,38]. Azari et al. (2013) reported two more cases of lethal head injury, in which a unilateral acute SDH and extensive optic nerve sheath hemorrhage were the results of a witnessed hard and repeated shaking as the primary physical mechanism of the brain and ophthalmic injuries [39]. One more case has been recently reported, exploring the injury biomechanics of SDS [40]. In a case of domestic violence, a young woman suffered post-concussional symptoms (including SDH, retinal hemorrhages and patterned bruising) for two weeks but survived. In this case, however, there was also evidence of a direct impact on the craniocervical region [41].

As above mentioned, rapid deceleration/ acceleration of the head or rapid changes in rotation and angular forces can easily occur during a violent shaking, resulting in tears of the bridging veins that drain blood into the dural venous sinuses [30,31] in adults too—especially if elderly. The intracranial changes that occur with aging can also be considered favorable factors for intracranial bleeding and the formation of SDH, even after mild or minor head trauma, with or without impact [32]. These age-related changes are mostly represented by increased adherence of the dura to the skull, cerebrovascular atherosclerosis increasing the bridging vein fragility, and progressive brain atrophy extending the distance between the cortex and the overlying coverings. This is the reason why the minimum force required to produce serious intracranial consequences is still not known. It has been reported that about 30% of all cases of SDH involve isolated SDH without associated skull fracture, skin wounds and cortical or intracranial hemorrhages [34,42]. The absence of external signs of scalp lacerations, skull fractures, or soft tissue swelling in many cases of SDH probably results in the role of the impact injury being underestimated. In our case, there were signs suspicious of multiple impact injuries to the head, as represented by the multiple small bruises on the vertex and the face (mostly at the left side), associated with a frontoparietal hematoma and a bilateral subacute SDH. The patterned polychromatic bruises distributed on the dorsal and lateral surfaces of the upper extremities and lateral chest on both sides were instead consistent with forceful grasping/gripping or repeated blows. Since SDH can be caused by rapid acceleration/rotation in absence of impact or by mild/minor blows, it is such a mechanism that probably occurred in our case. A number of underlying conditions may predispose one to SDH, including coagulopathies and anticoagulant therapy. In our case, the use of an antiplatelet drug aggregation inhibitor (ticlopidine) was also an additional factor increasing the risk of intracranial bleeding. In fact, the iatrogenic etiology of SDH was taken into account but considered as only a contributing factor and not a primary cause of death. Ticlopidine is an adenosine diphosphate (ADP) receptor inhibitor that interferes with platelet membrane function and platelet–platelet interactions, reducing the risk of thrombotic strokes. As ticlopidine prolongs bleeding time, its use is not indicated in cases of bleeding diathesis, hematological and sever liver diseases or in association with drugs that increase the risk of bleeding such as non-steroidal anti-inflammatory drugs (NSAIDs) or anticoagulant agents. Apart from ecchymosis, epistaxis, hematuria and conjunctival hemorrhage, among the adverse reactions, gastrointestinal rather than intracerebral bleeding is also described [43,44]. Neither contraindications nor overdosage were observed in the present case and the victim had been properly taking ticlopidine for many years without any adverse effect. Spontaneous SDH exists, although its pathogenesis has been questioned [34,45,46]. Most spontaneous SDHs are arterial in origin, caused by hypertensive peaks in blood pressure or from the rupture of a cerebral arterial aneurysm or even related to hematological or hepatic coagulation disorders. However, in the present case, no elements suggestive of a non-traumatic cause of SDH were found at autopsy, including aneurysm or arterial–venous malformation rupture, cocaine abuse, metastatic cancer or Moyamoya disease [45,47,48]. The results of the CT-scan performed two months before the fatal event ruled out intracranial hemorrhages and other main pathological abnormalities, only detecting signs of age-related intracranial changes, as represented by cerebral atrophy with enlargement of the intracranial cerebrospinal fluid spaces and chronic ischemic encephalopathy (Figure 1A,B). An additional point worthy of consideration is the following: SDHs are often seen in falls. Unfortunately, age is one of the key risk factors for falls and older people have the highest risk of death or serious injury, with devastating consequences arising from a fall. It is estimated that more than one-third of persons 65 years of age or older fall each year. Falls represent a public health problem since they are the second leading cause of unintentional injury deaths worldwide [4]. However, in our case, the association of bilateral SDH with multiple polychromatic bruises on the face, the chest and the upper limbs were more consistent with repeated physical abuses than a single accidental fall. This assessment was central to the subsequent judicial process; following investigations proved that physical abuse occurred.

In the discrimination of falls versus blows, the hat brim rule can be a moderately valid tool—especially if associated with other criteria [49]. According to these criteria, a higher number of facial contusions are in favor of blows, although facial injuries are not very useful alone in pointing toward one circumstance of death or another.

## 4. Conclusions

Elder abuse is associated with significant morbidity and mortality. Although SDH is commonly the result of a TBI, it can be also related to non-traumatic intracranial bleeding. Therefore, the traumatic and non-traumatic origin of SDH is still a challenge for forensic pathologists. As largely accepted in the pediatric population and also occasionally described in adults, violent shaking should be also considered a possible mechanism of SDH—especially in elderly individuals who do not have any sign of impact to the head. Intracranial age-related changes and the common use of anticoagulant/antiplatelet medications can be factors contributing to tears in bridging veins and space-occupying hemorrhages such as SDH, even after a minor head trauma. The possibility of a minor or occult head impact should be always considered in elderly people with SDH, as in the present case. However, according to the main biomechanical aspects of SDH, non-contact mechanisms such as violent shaking can also cause intracranial bleeding, and they should be taken into account when no skin wounds or skull fractures are present. The opportunity to perform an accurate death investigation is strongly recommended, as it should necessarily proceed with a forensic autopsy [50,51].

## Figures and Tables

**Figure 1 healthcare-11-00228-f001:**
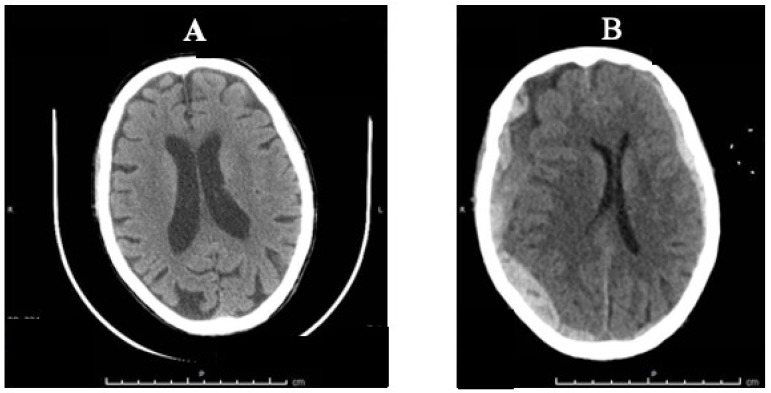
CT-scan comparison: (**A**) Two-month-old CT scan with signs of chronic ischemic encephalopathy and enlargement of intracranial cerebrospinal fluid spaces; (**B**) CT-scan made in the hospital showing bilateral SDH.

**Figure 2 healthcare-11-00228-f002:**
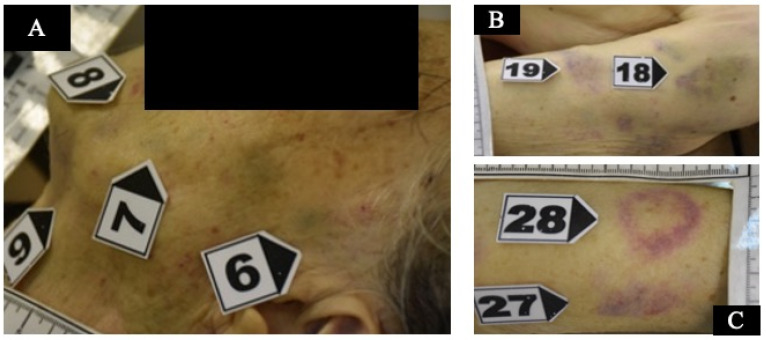
(**A**) bruises on the left side of the face; (**B**) bruises on the upper left arm; (**C**) bruises on the upper right arm.

**Figure 3 healthcare-11-00228-f003:**
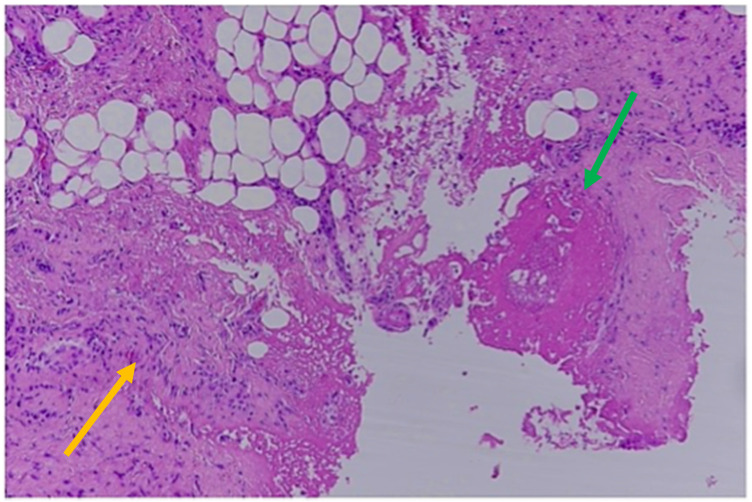
Histological findings of 5-day-old bruises: new capillary blood vessels (green arrow); granulation tissue; infiltration of macrophages and lymphocytes (yellow arrow; Hematoxylin & Eosin).

**Figure 4 healthcare-11-00228-f004:**
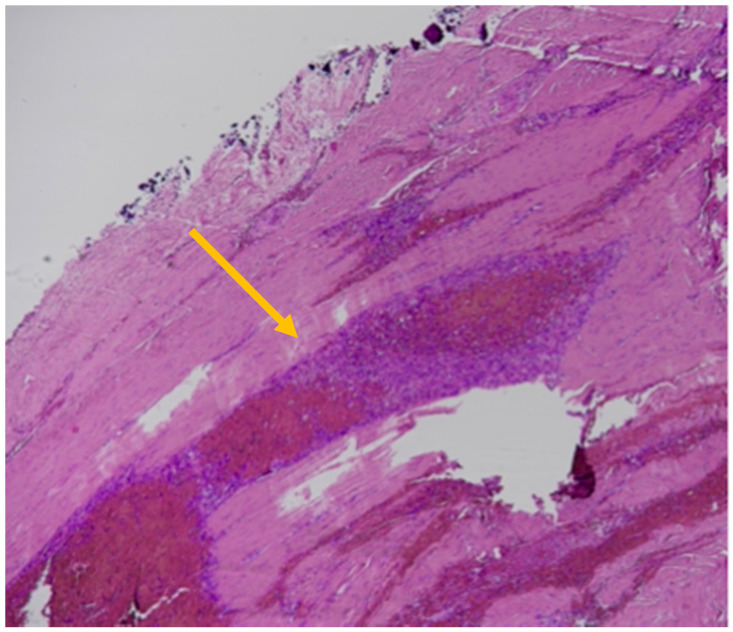
Histological findings of subacute SDHs > 5 days old: thin layer of fibroblasts between the dura and the clot (yellow arrow); granulation tissue; siderophages; breakdown of erythrocytes (yellow arrow; Hematoxylin & Eosin).

## Data Availability

The data presented in this study are available on request from the corresponding author. The data are not publicly available due to privacy restriction.

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
