# Peer review of "Accidental Injury or “Shaken Elderly Syndrome”? Insights from a Case Report"

_healthcare, 2023, doi:10.3390/healthcare11020228_

Round 1

Reviewer 1 Report

Thank you for giving me the opportunity to review the manuscript entitled “Accidental injury or “shaken elderly syndrome”? Insights from a case report”. The topic is relevant and is well-justified. The authors reported an autopsy case of an elderly woman's death caused by traumatic brain injury with multiple polychromatic bruises and subdural haemorrhage. The paper is well-written and discussed the challenging points for forensic pathologists. I think it is suitable for publication in this Journal.

Only minor revisions are suggested:

In Figure 3 and Figure 4, can authors mark what described to be observed in the figures for the clearer understanding for readers in different forensic fields.

Author Response

To Reviewer’s #1 comments:

R1: Thank you for giving me the opportunity to review the manuscript entitled “Accidental injury or “shaken elderly syndrome”? Insights from a case report”. The topic is relevant and is well-justified. The authors reported an autopsy case of an elderly woman's death caused by traumatic brain injury with multiple polychromatic bruises and subdural haemorrhage. The paper is well-written and discussed the challenging points for forensic pathologists. I think it is suitable for publication in this Journal.

Only minor revisions are suggested:

In Figure 3 and Figure 4, can authors mark what described to be observed in the figures for the clearer understanding for readers in different forensic fields.

A1: Thank you very much for this comment. We are glad not only that the reviewer has appreciated the topic of our manuscript but also the challenging points for forensic pathologists.

About suggestion authors put colored arrows on the figures to better showing histological findings also for non-pathologist as well.

Reviewer 2 Report

Dear authors,

I have read your manuscript and have left comments throughout (please see attached form). Apart from those specific comments, some references used for demographic data are outdated, which may change the argument and data presentation. If no updates exist, this should be stated. 

I also have a major concern regarding the use of the data and consent. Although informed consent was waived, following legislation cited, the authors are highly encouraged to provide and self-assessment ethical statement. To access data from a patient and have it published information with no consent needs to be addressed - ethical issues are a significant concern and should be addressed appropriately.   Also, note that according to the legislation mentioned:

_A clinical trial shall be subject to scientific and ethical review (article 4)

_ Also, informed consent is never waived but conditioned, especially if the patient cannot give consent - PLEASE address this point.

Also critical - this case study is not the result of a clinical trial - hence additional care is necessary when processing and using the information.

Author Response

To Reviewer’s #2 comments:

Dear authors,

R1: I have read your manuscript and have left comments throughout (please see attached form). Apart from those specific comments, some references used for demographic data are outdated, which may change the argument and data presentation. If no updates exist, this should be stated. 

A1: Thank you very much for this comment. We inserted in the manuscript new data and modify some sentences. Please check the INTRODUCTION where we modify not only the specific comments, but we rephrased the starting paragraph.

R2: I also have a major concern regarding the use of the data and consent. Although informed consent was waived, following legislation cited, the authors are highly encouraged to provide and self-assessment ethical statement. To access data from a patient and have it published information with no consent needs to be addressed - ethical issues are a significant concern and should be addressed appropriately.   Also, note that according to the legislation mentioned:

A clinical trial shall be subject to scientific and ethical review (article 4)

Also, informed consent is never waived but conditioned, especially if the patient cannot give consent - PLEASE address this point.

Also critical - this case study is not the result of a clinical trial - hence additional care is necessary when processing and using the information.

We specify that the report submitted to the journal, in accordance with national and European legislation n. 536/2014 and in accordance with the Guidelines for information on judicial data, G.U. Italian n. 24/01/2011 for the privacy of the deceased patient, does not require the prior opinion of an Ethics Committee or other authorizations as it does not refer to a drug trial and the case report has been completely anonymized and made not recognizable.

Round 2

Reviewer 2 Report

Dear Authors,

Thank you for addressing most of the comments and providing additional information, content, and references.

However, I continue to urge the authors to provide and self-assessment ethical statement.

Access and use of human remains and data on human patients must be acknowledged and adequately addressed - without the patient data, the research could not be carried out. Hence, acknowledgment is essential. Also, the need to secure consent on data use. IF not from the patient, from living relatives, and/or tutors. Ethical behavior needs to be considered all time, alongside legislation.

To access data from a patient and have it published with no consent needs to be addressed.

This practice must be incorporated into the research protocols of all of us undertaking research with humans (to say the least). 

And I would like to stress, again, that this case study is not the result of a clinical trial - so I need to take care when processing and using the information.